# Pleiotropic Effects of *APOB* Variants on Lipid Profiles, Metabolic Syndrome, and the Risk of Diabetes Mellitus

**DOI:** 10.3390/ijms232314963

**Published:** 2022-11-29

**Authors:** Shih-Jung Jang, Wei-Lun Tuan, Lung-An Hsu, Leay-Kiaw Er, Ming-Sheng Teng, Semon Wu, Yu-Lin Ko

**Affiliations:** 1Cardiovascular Center and Division of Cardiology, Department of Internal Medicine, Taipei Tzu Chi Hospital, Buddhist Tzu Chi Medical Foundation, New Taipei City 23142, Taiwan; 2School of Medicine, Tzu Chi University, Hualien 97004, Taiwan; 3Department of Internal Medicine, Taipei Tzu Chi Hospital, Buddhist Tzu Chi Medical Foundation, New Taipei City 23142, Taiwan; 4The First Cardiovascular Division, Department of Internal Medicine, Chang Gung Memorial Hospital and Chang Gung University College of Medicine, Taoyuan 33305, Taiwan; 5The Division of Endocrinology and Metabolism, Department of Internal Medicine, Taipei Tzu Chi Hospital, Buddhist Tzu Chi Medical Foundation, New Taipei City 23142, Taiwan; 6Department of Research, Taipei Tzu Chi Hospital, Buddhist Tzu Chi Medical Foundation, New Taipei City 23142, Taiwan; 7Department of Life Science, Chinese Culture University, Taipei 11114, Taiwan

**Keywords:** apolipoprotein B, *APOB*, Asian specific mutations, lipid profile, metabolic syndrome, Mendelian randomization, diabetes mellitus

## Abstract

Apolipoprotein B (ApoB) plays a crucial role in lipid and lipoprotein metabolism. The effects of *APOB* locus variants on lipid profiles, metabolic syndrome, and the risk of diabetes mellitus (DM) in Asian populations are unclear. We included 1478 Taiwan Biobank participants with whole-genome sequence (WGS) data and 115,088 TWB participants with Axiom genome-wide CHB array data and subjected them to genotype–phenotype analyses using *APOB* locus variants. Five *APOB* nonsynonymous mutations, including Asian-specific rs144467873 and rs13306194 variants, were selected from participants with the WGS data. Using a combination of regional association studies, a linkage disequilibrium map, and multivariate analysis, we revealed that the *APOB* locus variants rs144467873, rs13306194, and rs1367117 were independently associated with total, low-density lipoprotein (LDL), and non-high-density lipoprotein (non-HDL) cholesterol levels; rs1318006 was associated with HDL cholesterol levels; rs13306194 and rs35131127 were associated with serum triglyceride levels; rs144467873, rs13306194, rs56213756, and rs679899 were associated with remnant cholesterol levels; and rs144467873 and rs4665709 were associated with metabolic syndrome. Mendelian randomization (MR) analyses conducted using weighted genetic risk scores from three or two LDL-cholesterol-level-associated *APOB* variants revealed significant association with prevalent DM (*p* = 0.0029 and 8.2 × 10^−5^, respectively), which became insignificant after adjustment for LDL-C levels. In conclusion, these results indicate that common and rare *APOB* variants are independently associated with various lipid levels and metabolic syndrome in Taiwanese individuals. MR analyses supported *APOB* variants associated with the risk of DM through their associations with LDL cholesterol levels.

## 1. Introduction

Apolipoprotein B (apoB) is a critical component of all atherogenic lipoproteins. It maintains the structural stability of lipoproteins and plays a crucial role in lipid metabolism [1,2,3,4]. The retention and modification of apoB- and atherogenic-cholesterol-containing remnant lipoproteins within the arterial wall constitute a fundamental step in atherosclerosis, which involves initial fatty streak deposition, followed by progression to complex atherosclerotic lesions that are vulnerable to plaque rupture or endothelial erosion [5]. Each atherogenic lipoprotein particle precisely contains a single molecule of apoB with two major isoforms: apoB48 for chylomicron and chylomicron remnants and apoB100 for lipoprotein (a), low-density lipoprotein (LDL), intermediate-density lipoprotein (IDL), and very-LDL (VLDL) [1]. Most VLDL particles approximate LDL particles for the risk of atherogenesis. Thus, the plasma concentration of apoB can be derived as the total number of apoB particles; this concentration can serve as a direct measure of the number of circulating atherogenic lipoproteins and of the total atherogenic risk attributable to both VLDL and LDL particles [4]. For the prediction of cardiovascular outcomes, most clinical trials have reported apoB levels to be superior to LDL cholesterol levels; however, not all studies have concluded that apoB is superior to non-high-density lipoprotein (non-HDL) cholesterol levels [4,6,7]. Therefore, circulating apoB levels constitute a stronger, or at least an equivalent, predictor of future cardiovascular events when compared with LDL and non-HDL cholesterol levels. Furthermore, other studies have identified that apoB levels with plasma lipids could further improve risk prediction and management for patients with atherogenic dyslipoproteinemia [8].

The *APOB* gene, located on chromosome 2p24.1, contains 29 exons and has a total length of 43 kb [9]. *APOB* variants have been extensively studied over the past three decades and more than 5000 polymorphic sites have been reported at the highly polymorphic *APOB* locus (https://www.ncbi.nlm.nih.gov/snp/?term=APOB, accessed on 25 June 2022), which may be associated with familial hypercholesterolemia or hypobetalipoproteinemia (HBLP) [10,11,12,13]. However, the effects of *APOB* variants on lipid profiles, metabolic syndrome, and diabetes mellitus (DM) in Asian populations have not been fully elucidated. During evolutionary history, populations in different geographic areas may have experienced genetic drift and evolutionary selection, leading to ethnic heterogeneity in genetic architectures [14]. Accordingly, we conducted the present study to test the effects of *APOB* variants on lipid profiles, metabolic syndrome, and the risk of DM in Taiwanese individuals by applying the candidate variant approach and regional association analysis to the data of over 110,000 individuals selected from the Taiwan Biobank (TWB) [15,16]. Our results revealed a significant association between common and rare *APOB* variants with multiple lipid levels and metabolic syndrome. Further, Mendelian randomization (MR) analyses provide supportive evidence that *APOB* variants are associated with the risk of DM through their associations with LDL cholesterol levels. The results thus revealed the crucial role of ethnic-specific *APOB* variants in predicting cardiometabolic disorders in Taiwanese individuals. Our findings also highlight the importance of individually searching for ethnic-specific variants in each study population in different geographic areas.

## 2. Results

### 2.1. Selection of Candidate Nonsynonymous Mutations of the APOB Gene

Figure 1 showed the flowchart of inclusion and exclusion criteria used to screen TWB project participants. Table 1 lists the baseline characteristics, lipid profiles, metabolic syndrome, and DM of the participants selected from the TWB, including 115,088 participants with Axiom genome-wide CHB array data and 1478 participants with WGS data. Previous studies have revealed ethnic-specific *APOB* exonic mutations associated with LDL cholesterol levels, familial hypercholesterolemia, and HBLP [10,11,12,17,18,19]. We selected candidate *APOB* nonsynonymous mutations from our WGS data. We identified a total of five mutations: rs144467873 (p.Arg3527Trp), a rare variant; and rs676210 (p.Pro2739Leu), rs679899 (p.Ala618Val), rs13306194 (p.Arg532Trp), and rs1367117 (p.Thr98Ile), all common variants (Appendix A). A preliminary analysis of the WGS data of the 1478 participants using regional association analysis revealed that rs144467873 was the lead SNP associated with LDL cholesterol levels (*p* = 7.75 × 10^−5^, Appendix A). In the five nonsynonymous mutations, rs144467873 was also the only variant associated with LDL cholesterol levels (Appendix A).

### 2.2. Genotype–Phenotype Association Analysis of APOB Nonsynonymous Mutations with Lipid Profiles and Metabolic Syndrome

We used genome-wide CHB array data to test the association of the identified *APOB* mutations with lipid profiles and metabolic syndrome in the TWB participants. Significant associations were noted between all five mutations and the total, LDL, and non-HDL cholesterol levels. Additionally, different degrees of association were noted between the *APOB* mutations and various other lipid levels, including triglyceride and HDL and remnant cholesterol levels (Table 2). Significant associations were also noted between the rs144467873, rs679899, and rs13306194 genotypes and metabolic syndrome (*p* = 2.58 × 10^−8^, *p* = 0.0016, and *p* = 1.40 × 10^−5^, respectively).

### 2.3. Regional Association Studies with Conditional Analyses for the Association of APOB Locus Variants with Lipid Profiles and Metabolic Syndrome

Regional association studies along with conditional analyses were performed to determine the association between genetic variants near the *APOB* gene region and the study phenotypes. Our results revealed that the lead SNPs for each phenotype were located at or near the *APOB* region, indicating pleiotropic effects on this gene locus (Figure 2). Independent genome-wide significant associations were observed between the rs13306194 and rs7575840 variants and total cholesterol levels, between the rs13306194 and rs17399144 variants and LDL cholesterol levels, between the rs13306194 and rs7575840 variants and non-HDL cholesterol levels, between the rs1318006 variant and HDL cholesterol levels, between the rs35131127 and rs13306194 variants and triglyceride levels, and between the rs13306194 and rs56213756 variants and remnant cholesterol levels. We further performed a regional association analysis to test the association between the *APOB* variants and metabolic syndrome. The lead SNP was rs4665709 (*p* = 3.24 × 10^−8^), situated at approximately 3 kb downstream of *APOB*.

### 2.4. LD between APOB Nonsynonymous Mutations and Lead SNPs

We tested the LD between the *APOB* nonsynonymous mutations and lead SNPs (Figure 3). Our results revealed strong LD between rs7575840, rs17399144, and rs1367117 (r^2^ = 0.854–0.956) and between rs56213756 and rs1318006 (r^2^ = 0.985). Moderate LD was observed between four downstream lead SNPs, namely rs35131127, rs56213756, rs1318006, and rs4665709 (r^2^ = 0.282–0.702). Moderate LD was also noted between rs676210 and rs13306194 and four downstream lead SNPs (r^2^ = 0.202–0.421), and between rs679899 and rs7575840, rs17399144, rs1367117, and rs693 (r^2^ = 0.243–0.576). Furthermore, all downstream lead SNPs were in weak LD (r^2^ < 0.200) with the five nonsynonymous mutations. The rare exonic mutation rs144467873 was also in weak LD with all other *APOB* SNPs.

### 2.5. Genotype–Phenotype Association Analysis of Lead SNPs Downstream of the APOB Gene with Lipid Profiles and Metabolic Syndrome

We tested the association of the lead SNPs downstream of *APOB* with lipid profiles and metabolic syndrome (Appendix A). Our results revealed that almost all lead SNPs had significant associations with various lipid profiles; however, compared with the exonic mutations, the downstream *APOB* variants had weaker effects on total, LDL, and non-HDL cholesterol levels but stronger effects on triglyceride and remnant cholesterol levels. All four lead SNPs downstream of *APOB* also had strong association with metabolic syndrome (all *p* < 6.0 × 10^−8^).

### 2.6. Stepwise Linear Regression Analysis for Lipid Profiles

Owing to strong LD between the lead SNPs rs7575840 and rs17399144 and the nonsynonymous mutation rs1367117, we used rs1367117 to represent rs7575840 and rs17399144 in the multivariate analysis of the genetic determinants of lipid profiles. We used a threshold of genome-wide significance in initial genotype-phenotype association results for further multivariate analysis. For those variants with at least moderate LD (r^2^ > 0.3), we included only those variants with the strongest significance for multivariate analysis. Thus, for total LDL and non-HDL cholesterol levels, a total of three variants were used for multivariate analysis. Further, for triglyceride, HDL and remnant cholesterol levels, a total of one, two, and four variants, respectively, were used for multivariate analysis (Table 3). The stepwise linear regression analysis results showed independent associations of rs13306194, rs1367117, and rs144467873 with total, LDL, and non-HDL cholesterol levels; rs1318006 with HDL cholesterol levels; rs144467873, rs13306194, rs56213756, and rs679899 with remnant cholesterol levels; and rs35131127 and rs13306194 with triglyceride levels. These accounted for 0.82%, 0.89%, 0.91%, 0.08%, 0.21%, and 0.17% of the variance in total cholesterol, LDL cholesterol, non-HDL cholesterol, HDL cholesterol, remnant cholesterol, and triglyceride levels, respectively (Table 3).

### 2.7. Logistic Regression Analysis for Metabolic Syndrome

We conducted a logistic regression analysis using age, sex, body mass index, current smoking, and two *APOB* variants. Due to at least moderate LD between the lead SNPs downstream of *APOB*, we selected rs4665709, the lead SNP of metabolic syndrome, for analysis. The results demonstrated that rs144467873 and rs4665709 were independently associated with the risk of metabolic syndrome (odds ratio = 1.93 and 1.13, 95% CI = 1.54–2.42 and 1.08–1.17, *p* = 1.60 × 10^−8^ and 2.16 × 10^−8^, respectively; Table 4).

### 2.8. MR Analysis for the APOB Variants and WGRSs for Causal Relationship between LDL Cholesterol Levels and DM

We performed two stage least square (2SLS) instrumental variable (IV) analysis to determine the direction and causality of the association between LDL cholesterol levels and DM status (Table 5). Three aforementioned *APOB* variants independently associated with LDL cholesterol levels were analyzed. When all three variants were analyzed, significant association with the prevalence of DM status was noted (*p* = 0.0340). The association of *APOB*-WGRS with DM status remained significant after adjustment for multiple parameters associated with LDL cholesterol levels (*p* = 0.0029), and the association became insignificant after adjustment for LDL cholesterol levels (*p* = 0.7509). Due to the highly pleiotropic effect of rs144467873, which includes the risk of metabolic syndrome, we repeated 2SLS using only the rs13306194 and rs1367117 for analysis. The WGRS of the two variants showed significant association with the prevalence of DM status (*p* = 0.0040), which became more significant after adjustment for multiple parameters (*p* = 8.2 × 10^−5^). The associations subsided after further adjustment of LDL cholesterol levels. The *F* statistics derived for the instruments ranged above 1000 for LDL-C-level-determining genotypes, demonstrating a low risk of weak instrument bias.

## 3. Discussion

This study clarified the effects of common and rare *APOB* variants on lipid profiles and metabolic syndrome in Taiwanese individuals (Figure 3). Our results revealed that three nonsynonymous mutations, including the two Asian-specific variants, rs144467873 and rs13306194, were independently associated with either elevated (rs1367117 and rs144467873) or decreased (rs13306194) total, LDL, and non-HDL cholesterol levels, respectively. Furthermore, the rs144467873 variant was independently associated with decreased remnant cholesterol levels and with an increased risk of metabolic syndrome. The rs13306194 variant was independently associated with reduced triglyceride and remnant cholesterol levels and the rs679899 variant was independently associated with remnant cholesterol level. In addition, *APOB* variants at the 3′ intergenic region played a crucial role in increased triglyceride and remnant cholesterol levels and decreased HDL cholesterol levels; they also increased the risk of metabolic syndrome. These results provide the evidence that both coding- and non-coding-region *APOB* variants are involved in lipid profiles and metabolic syndrome. We also noted that the *APOB* nonsynonymous mutations and downstream variants had differential effects on lipid profiles. In addition, MR analyses supported *APOB* variants associated with the risk of DM through their associations with LDL-C levels. Accordingly, scholars should explore ethnic-specific *APOB* variants as genetic determinants of cardiometabolic traits and metabolic syndrome in a specific population.

### 3.1. Ethnic-Specific APOB Variants for Total, LDL, and Non-HDL Cholesterol Levels in Taiwanese Individuals

Both rs144467873 (R3500W mutation) and rs13306194 are Asian-specific *APOB* variants [10]. The *APOB* R3500W and R3500Q mutations are the major causes of autosomal dominant hypercholesterolemia because of familial defective apolipoprotein B-100 (FDB) [10,12,20,21]. FDB-related hypercholesterolemia is characterized by elevated LDL cholesterol levels engendered by the reduced binding of LDL to LDL receptors and decreased clearance of apoB-containing particles by hepatocytes owing to decreased affinity between mutated *APOB*-100 and LDL receptors [10]. The elevation of LDL cholesterol levels caused by FDB has been reported to be relatively mild when compared with elevations caused by LDL receptor mutations [22,23,24]. Therefore, hypercholesterolemia in FDB is easily underdiagnosed. In Caucasians, the R3500Q mutation is the most common cause of FDB, with a frequency of approximately 0.1%; but the R3500W mutation is likely rare [25,26,27]. However, a study including 373 patients with hyperlipidemia in Taiwan reported that the prevalence of heterozygote R3500Q was 0.3% and that the prevalence of heterozygote R3500W was 2.4% [12]; these results suggest that R3500W, rather than R3500Q, could be the principal mutation responsible for FDB in Taiwanese individuals. Yang, et al. [21] analyzed 87 patients with autosomal-dominant hypercholesterolemia who were selected from 30 unrelated Taiwanese families; they observed that three families (10%) had the *APOB* R3500W mutation. Huang, et al. [11] analyzed 750 index and unrelated familial patients with hypercholesterolemia and found that 58 patients (7.7%) had the *APOB* R3500W mutation. Using the data of the 115,088 TWB participants for analysis, we found 354 heterozygous R3500W mutations and one homozygous R3500W mutation, with the corresponding minor allele frequency (MAF) being 0.16%. The very high LDL cholesterol level (207 mg/dL, comparing to mean 163 mg/dL for participants with heterozygous R3500W mutation) of the participant with homozygous R3500W mutation also suggested the co-dominant inheritance of this variant. These results indicate that the *APOB* rs144467873 variant plays a critical role in determining LDL cholesterol levels in Taiwanese individuals.

The rs13366194 variant, which was a common *APOB* mutation in our study population, was also noted to be independently associated with total, LDL, and non-HDL cholesterol levels. This variant was first reported by Tang, et al. [19] in Chinese individuals, with its MAF being 0.13. Two Korean studies have subsequently found this variant in patients with extremely low LDL cholesterol levels and in Korean Biobank participants [17,18]. The MAF of rs13366194 was 0.14 in our study participants, similar to those reported in Chinese and Korean populations. The MAF of rs13366194 is extremely low in 1000 Genome European (0.3%) and African (0.04%) samples and in HUNT-MI Norwegians (0.07%), according to the PUBMed website (PUBMed.gov), indicating that rs13366194 is another Asian-specific variant. Previous genome-wide association studies (GWASs) have demonstrated associations of rs1367117 genotypes with total cholesterol, LDL cholesterol, HDL cholesterol, and triglyceride levels and with metabolic syndrome [28,29,30,31]. In our study population, the rs1367117 genotypes (MAF = 0.13) were observed to be associated with total, LDL, and non-HDL cholesterol levels; the MAF of these genotypes is lower than that in European populations (approximately 0.30) but is equal to those in African and other Asian populations (approximately 0.11–0.12), according to the PUBMed website (PUBMed.gov). Different MAFs of rs676210 and rs679899 were also noted in ethnic distinct populations as shown in the PUBMed website (PUBMed.gov). Previous GWAS and genetic association studies have shown significant association between rs676210 genotypes and apolipoprotein A1, HDL cholesterol and triglyceride and oxidized LDL levels as well as serum metabolite measurement and the response to fenofibrate therapy [32,33,34,35,36,37,38], and between rs679899 genotypes and HDL and LDL cholesterol and triglyceride levels and chronic kidney disease in hypertension [39,40,41].

### 3.2. Variants in the 3′ Intergenic Region of APOB as Genetic Determinants of Triglyceride and HDL and Remnant Cholesterol Levels

We found that the downstream *APOB* lead variants were independently associated with triglyceride and HDL and remnant cholesterol levels. Specifically, we noted that all four *APOB* downstream lead variants, namely rs35131127, rs56213756, rs1318006, and rs4665709, were in weak LD with various *APOB* nonsynonymous mutations and upstream *APOB* region variants. None of the four downstream variants have been reported in the literature. The downstream variants may play a regulatory role in the expression of *APOB* with microRNA binding as an important mechanism [42]. The plasma apoB concentration can serve as a measure of the number of circulating atherogenic lipoproteins. Moreover, apoB-containing remnant lipoproteins deposited in the vessel wall are involved in the development and progression of atherosclerosis [5]. Further larger prospective studies with trans-ethnic populations and meta-analyses will be necessary to clarify more of the effects of *APOB* downstream variants on cardiometabolic traits.

### 3.3. APOB Variants as Genetic Determinants of Remnant Cholesterol Levels

Remnant cholesterol refers to the cholesterol content of VLDLs and IDLs in the fasting state and to the combined cholesterol content of VLDLs, IDLs, and chylomicrons in the non-fasting state [43,44]. Apart from LDL cholesterol and apoB levels, remnant cholesterol is a causal risk factor and a predictor of atherosclerotic cardiovascular disease [45,46]. MR studies have indicated that elevated remnant cholesterol levels are associated with 25-hydroxyvitamin D deficiency and aortic valve stenosis [47,48]. A GWAS study determined that *APOB* rs693 genotypes were associated with remnant cholesterol levels [37]. Our results reveal that rs693 genotypes were significantly associated with all lipid profile components; however, in the multivariate analysis, the degree of association of these genotypes with the lipid profile components decreased, except for HDL cholesterol levels (Appendix A). In our study population, in addition to the rs56213756 variant in the 3′ intergenic region of *APOB*, both Asian-specific variants rs144467873 and rs13306194 as well as rs679899 were independently associated with remnant cholesterol levels. These results indicate the importance of ethnic heterogeneity in the genetic determination of remnant cholesterol levels using *APOB* variants.

### 3.4. Novel APOB Variants Determining Metabolic Syndrom

In this study, two novel candidate variants, namely rs4665709 and rs144467873, were associated with an increased risk of metabolic syndrome (by 1.13- and 1.93-fold, respectively). Metabolic syndrome is a complex disorder that is characterized using body shape indices, such as the waist–hip ratio, and metabolic factors, such as hypertension, dyslipidemia, and impaired glucose tolerance. Kristiansson, et al. [49] were the first to report the association of *APOB* rs673548 and rs6728178 with an increased risk of metabolic syndrome. Lind [50] also revealed that *APOB* rs673548 was associated with an increased risk of metabolic syndrome. Furthermore, Guindo-Martínez, et al. [31] demonstrated that rs1367117 genotypes were associated with metabolic syndrome. 

The aforementioned studies have all been conducted on Caucasian populations. In this study, we noted rs673548 to be in nearly complete LD with rs676210, a nonsynonymous mutation of *APOB* (Appendix A). Rs673548, rs6728178, and rs676210 were not associated with the risk of metabolic syndrome, and rs1367117 genotypes were not independently associated with metabolic syndrome in our population (Table 2 and Appendix A). In our study, rs4665709 was associated with HDL, non-HDL, and remnant cholesterol and triglyceride levels, which may partially explain the association of this variant with metabolic syndrome (Appendix A). By contrast, rs144467873 was not associated with any component of metabolic syndrome in our study population, despite being significantly associated with metabolic syndrome in the TWB participants (Table 2 and Appendix A). Our previous study revealed genetic risk of inflammation to be associated with metabolic syndrome [51], and enhanced inflammation was shown to be associated with familial hypercholesterolemia [52]. Lind [50] conducted a pathway analysis and reported that genetic loci related to the immune system, transportation of small molecules, and metabolism are associated with the risk of metabolic syndrome. Further research is necessary to elucidate the mechanism underlying the association of rs144467873 with the risk of metabolic syndrome.

### 3.5. MR Analyses

The cause-effect relationship between LDL cholesterol levels and DM has been suggested to be gene- or mechanism-specific. We have previously used seven common and rare *PCSK9* variants and 41 genetic variants from a GWAS for LDL cholesterol levels, including the *APOB* rs13306194, to test the causal relationship between LDL cholesterol levels and DM in 75,441 TWB participants [53]. In this study, with the extension of the number of TWB participants to more than 110,000, we further tested whether *APOB* variants were associated with the risk of DM through their associations with LDL cholesterol levels. Familial hypercholesterolemia, caused by mutations on the *LDLR, APOB,* and *PSCK9*, has shown a lower prevalence of DM when compared to their unaffected relatives [54]. Our data also revealed *APOB* variants and WGRSs showed supportive evidence of an inverse association between LDL cholesterol levels and DM. Further study with extension to other candidate genes for LDL cholesterol levels may help to elucidate more of the genetic pathways in determining the risk of DM.

### 3.6. Limitations

The limitations of this study are its cross-sectional design, the presence of ethnic-specific variants, and the absence of a second cohort. Therefore, future studies can conduct further replication using other study groups, especially ethnically distinct populations, to confirm our findings. Moreover, a larger prospective cohort study using WGS will be more powerful in detecting whether other functional variants are more critical genetic determinants of lipids profile and metabolic syndrome in the *APOB* locus.

## 4. Materials and Methods

### 4.1. TWB Cohort

The TWB is a population-based research database sponsored by the Taiwanese Government and compiled between 2008 and 2020. We selected the data of 129,542 participants who did not have a history of cancer and had Axiom genome-wide CHB array data. The data of 14,454 participants were excluded from the analysis in accordance with the following criteria: exhibiting cryptic relatedness (as determined by an identity-by-descent PI_HAT value of >0.187; n = 7216), fasting for less than 6 h (n = 4647), and missing genotyping data for any of the five *APOB* exonic mutations (rs13306194, rs1367117, rs144467873, rs676210, and rs679899; n = 2591). Figure 1 displays the participant enrollment flowchart. Individuals with a history of hyperlipidemia were excluded from the lipid profile analysis. The Appendix A presents the definition of metabolic syndrome and DM. We also performed ultrafast whole-genome secondary analysis on Illumina sequencing platforms to search for *APOB* nonsynonymous mutations among 1478 TWB participants who had whole-genome sequencing (WGS) data [55]. Ethical approval was granted by the Research Ethics Committee of Taipei Tzu Chi Hospital, Buddhist Tzu Chi Medical Foundation (approval number: 05-X04-007), and the Ethics and Governance Council of the TWB (approval number: TWBR10507-02 and TWBR10611-03). All participants provided written informed consent.

### 4.2. Clinical Phenotypes and Laboratory Examinations

In addition to baseline characteristics (i.e., age, sex, current smoking, and body mass index), we measured the lipid profiles (i.e., total cholesterol, LDL and HDL cholesterol, and triglyceride levels) by using colorimetric assays (Hitachi LST008, Automatic Clinical Chemistry Analyzer, Hitachi, Naka, Japan). Non-HDL and remnant cholesterol levels were calculated as total cholesterol minus HDL cholesterol levels and total cholesterol minus HDL cholesterol minus LDL cholesterol, respectively [56].

### 4.3. Regional Association Analysis for the WGS Data 

A total of 1478 WGS data from a subgroup of TWB participants were used for evaluation using an ultrafast whole-genome secondary analysis on Illumina sequencing platforms [55] (Illumina HiSeq 2500/4000). The resulting reads were aligned to the hg19 reference genome with iSAAC 01.13.10.21. Single nucleotide polymorphisms (SNPs) and insertion-deletion variant discovery and genotyping were analyzed by iSAAC Variant Caller 2.0.17 [55]. To combine 1478 vcf files, an in-house protocol written in shell script was performed. A union table of all detected variants between 21.32 kb and 21.36 kb in chromosome 2p24.1 was used for regional association analysis. The association between SNPs and LDL cholesterol levels was then analyzed using the GWAS method.

### 4.4. Regional Association Analysis for the GWAS Data

The lead SNPs for lipid profiles and metabolic syndrome near the *APOB* locus were identified through regional association analyses [57,58] using the Axiom genome-wide CHB array data of the participants after the application of the exclusion criteria (Figure 1). Using the 1000 Genomes Project Phase 3 East Asian populations as a reference panel, we performed genome-wide genotype imputation by employing SHAPEIT (version 2, Oxford, UK, https://mathgen.stats.ox.ac.uk/genetics_software/shapeit/shapeit.html, accessed on 2 December 2020) and IMPUTE2 (version 2, Oxford, UK, http://mathgen.stats.ox.ac.uk/impute/impute_v2.html, accessed on 2 December 2020). After the imputation, we conducted quality control assessments by filtering SNPs with IMPUTE2 imputation quality scores of >0.3 and removing insertion–deletion mutations by using VCFtools (version 0.1; https://vcftools.github.io/index.html, accessed on 2 December 2020). All samples included in the analyses had a call rate of ≥97%; moreover, among the SNPs included in subsequent analyses, the missing rate was <3%, MAF was ≥0.01, and *p* value for the violation of the Hardy–Weinberg equilibrium was ≥10^−6^. Finally, 115,088 participants were included in the regional association analysis for 400 SNPs near the *APOB* locus on chromosome 2p24.1 at positions ranging between 21.12 and 21.36 Mb.

### 4.5. Statistical Analysis

Data such as lipid profiles are presented as medians and interquartile ranges. Categorical data such as metabolic syndrome and DM are presented as percentages and numbers. Before the analyses, lipid profile parameters were transformed logarithmically for regression analysis. In the association studies, general linear regression was used to evaluate the effect of *APOB* region variants on lipid profiles after adjustment for age, sex, body mass index, and current smoking status. The effects of polymorphisms on the risk of metabolic syndrome and DM are expressed as odds ratios along with 95% confidence intervals (95% CIs); these effects were evaluated through logistic regression analysis. Moreover, we performed a stepwise linear regression analysis to determine the independent correlates of lipid profiles. We used a threshold of genome-wide significance in initial genotype-phenotype association results for further multivariate analysis. For those variants with at least moderate LD (r^2^ > 0.3), we included only those variants with the strongest significance for multivariate analysis. The regional association studies and conditional analyses were conducted using the PLINK software package (version 1.07; Shaun Purcell, Cambridge, MA, USA, https://zzz.bwh.harvard.edu/plink/, accessed on 14 August 2021). Genome-wide significance was defined as a *p* value of <5 × 10^−8^. LDmatrix (https://analysistools.nci.nih.gov/LDlink/?tab=ldmatrix, assessed on 19 April 2021) was used to analyze the LD. The data were analyzed using SPSS (version 22; SPSS, Chicago, IL, USA) software.

### 4.6. MR Analysis for the APOB Variants and WGRSs with the Risk of DM through Their Associations with LDL-C Levels

We further used 2SLS regression with IVs to examine whether *APOB* variants and WGRSs were associated with the risk of DM through their associations with LDL cholesterol levels. *APOB* variants and WGRSs were initially regressed to generate predicted LDL cholesterol levels during the first stage of the regression. The study parameters were then regressed on *APOB* variants and WGRSs to generate the predicted risk of DM during the second stage of the regression. To create WGRSs, the *APOB* variants were weighted according to each allele score by using the β coefficients from our GWAS analysis, and the risk allele exhibiting directionally concordant associations with the target parameters was selected. The F statistic was used to assess the strength of the instruments, and was calculated as previously reported [59]. An F statistic of >10 indicates a relatively low risk of weak instrumental bias in MR analyses [60]. 

## 5. Conclusions

Our results demonstrate the crucial role of *APOB* variants in various aspects of lipid profiles and metabolic syndrome. Nonsynonymous mutations, especially two Asian-specific mutations (rs144467873 and rs13366194), were noted to play a crucial role in determining LDL, non-HDL, and remnant cholesterol levels and metabolic syndrome. We also noted unique independent effects of *APOB* 3′ intergenic region variants on triglyceride levels, HDL and remnant cholesterol levels, and metabolic syndrome. MR analyses with *APOB* variants showed evidence of causally inverse association between LDL cholesterol levels and DM. Future prospective studies and meta-analyses, especially those including trans-ethnic populations, may help to verify our findings and clarify the influence of *APOB* variants on cardiometabolic traits and disorders. 

## Figures and Tables

**Figure 1 ijms-23-14963-f001:**
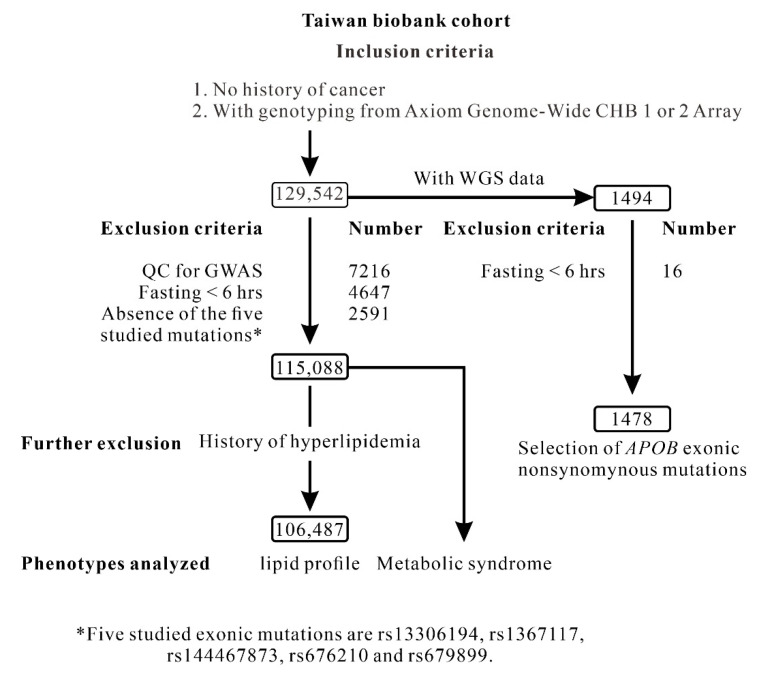
Flowchart of inclusion and exclusion criteria used to screen Taiwan Biobank project participants.

**Figure 2 ijms-23-14963-f002:**
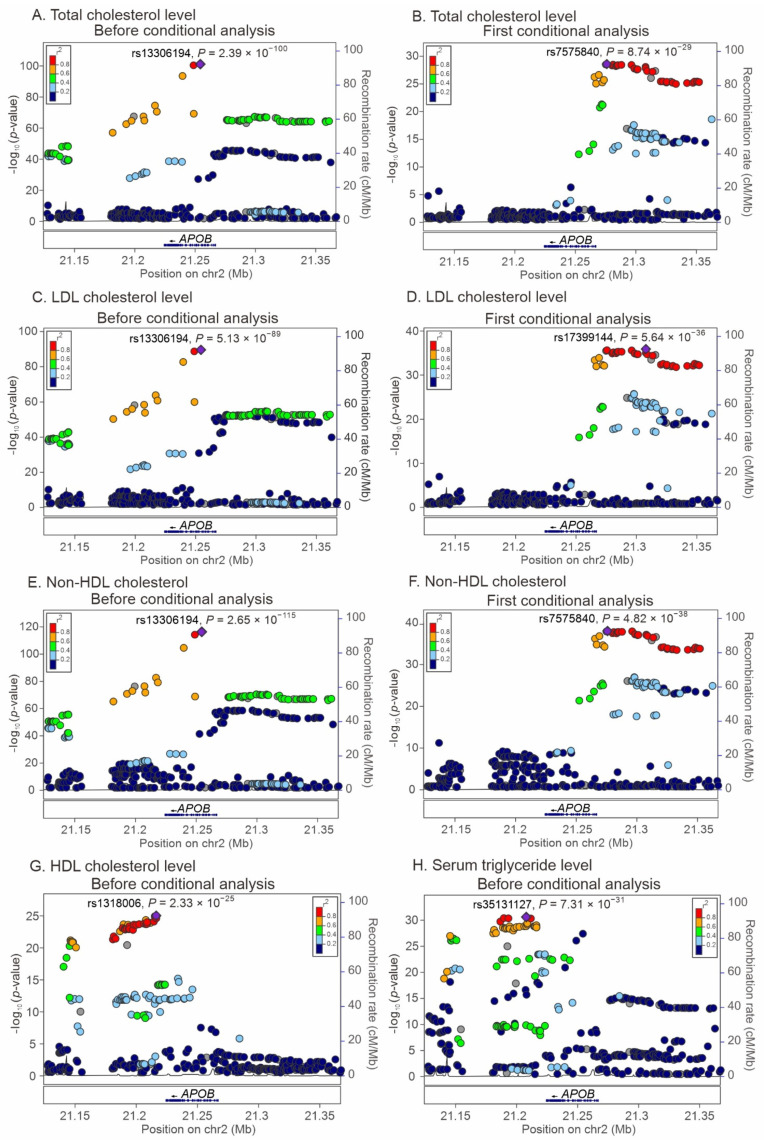
Regional association analysis between *APOB* variants and lipid profiles and metabolic syndrome.

**Figure 3 ijms-23-14963-f003:**
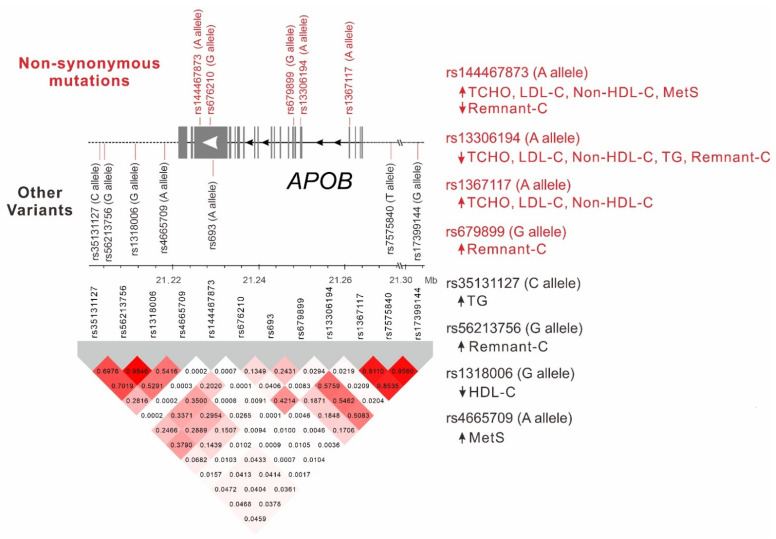
Linkage disequilibrium map of *APOB* region single-nucleotide polymorphisms (**left** panel) and summary of independent associations between *APOB* locus variants and lipid profiles and metabolic syndrome (**right** panel). The non-synonymous mutations were shown in red. Shades of red and gray show the strength of the pairwise linkage disequilibrium based on r^2^, and numbers indicate the value of r^2^.

**Table 1 ijms-23-14963-t001:** Baseline characteristics of Taiwan Biobank participants.

Clinical and Laboratory Parameters *	With WGS Data (*n* = 1478)	With GWAS Data (*n* = 115,088)
Anthropology		
Age (years)	50.0 (40.0–59.0) ^#^	51.0 (40.0–59.0) ^#^
Sex (male vs. female)	739/739	41,467/73,621
Body mass index (kg/m^2^)	23.9 (21.9–26.4)	23.8 (21.6–26.3)
Current smoking (%)	10.08% (149)	19.63% (22,590)
Lipid profile		
Total cholesterol (mg/dL)	191.0 (170.0–215.0)	193.0 (171.0–217.0)
LDL cholesterol (mg/dL)	119.0 (101.0–141.0)	119.0 (99.0–141.0)
Non-HDL cholesterol (mg/dL)	137.0 (116.0–160.0)	138.0 (116.0–162.0)
HDL cholesterol (mg/dL)	52.0 (44.0–63.0)	53.0 (45.0–63.0)
Triglyceride (mg/dL)	90.0 (64.0–130.0)	91.0 (64.0–133.0)
Remnant cholesterol (mg/dL)	16.0 (11.0–22.0)	16.0 (11.0–23.0)
Metabolic syndrome (%)	19.30% (285)	25.53% (29,384)
Diabetes mellitus (%)	9.07% (134)	9.45% (10,879)

WGS, whole-genome sequence; GWAS, genome-wide association study; HDL cholesterol, high-density lipoprotein cholesterol; LDL cholesterol, low-density lipoprotein cholesterol; non-HDL cholesterol, non-high-density lipoprotein cholesterol. * Participant selection criteria for analysis are displayed in Figure 1. ^#^ Level is presented as median (interquartile range).

**Table 2 ijms-23-14963-t002:** Association between *APOB* nonsynonymous mutations and serum lipid profiles and metabolic syndrome.

Genetic Variants	Genotypes			Beta	SE	*p* Value
*APOB* rs144467873	GG (106,145)	GA (341)	AA (1)			
Total cholesterol (mg/dL)	193.0 (171.0–217.0)	231.0 (206.0–255.5)	278.0	0.0805	0.0041	2.29 × 10^−85^
LDL cholesterol (mg/dL)	119.0 (99.0–140.0)	163.0 (138.0–188.0)	207.0	0.1373	0.0061	1.96 × 10^−110^
Non-HDL cholesterol (mg/dL)	138.0 (116.0–162.0)	176.0 (154.0–203.0)	238.0	0.1148	0.0055	6.08 × 10^−95^
HDL cholesterol (mg/dL)	53.0 (45.0–63.0)	53.0 (44.0–61.0)	40.0	−0.0124	0.0049	0.0114
Triglyceride (mg/dL)	91.0 (64.0–133.0)	82.0 (60.0–127.0)	150.0	−0.0077	0.0115	0.5037
Remnant cholesterol (mg/dL)	16.0 (12.0–23.0)	13.8 (8.0–19.0)	31.0	−0.1005	0.0142	1.71 × 10^−12^
Metabolic syndrome (%)	25.50% (29,235)	33.94% (149)	100% (1)	0.6463	0.1161	2.58 × 10^−8^
*APOB* rs676210	AA (56,252)	AG (42,386)	GG (7849)			
Total cholesterol (mg/dL)	194.0 (172.0–218.0)	192.0 (170.0–216.0)	190.0 (168.0–213.0)	−0.0047	0.0004	1.53 × 10^−36^
LDL cholesterol (mg/dL)	120.0 (100.0–142.0)	119.0 (99.0–139.0)	117.0 (98.0–137.0)	−0.0063	0.0006	4.10 × 10^−30^
Non-HDL cholesterol (mg/dL)	139.0 (117.0–163.0)	137.0 (116.0–161.0)	135.0 (114.0–158.0)	−0.0053	0.0005	1.03 × 10^−25^
HDL cholesterol (mg/dL)	54.0 (45.0–63.0)	53.0 (45.0–63.0)	53.0 (44.0–62.0)	−0.0031	0.0004	1.75 × 10^−12^
Triglyceride (mg/dL)	90.0 (63.0–134.0)	91.0 (64.0–133.0)	91.0 (66.0–133.0)	0.0021	0.0010	0.0424
Remnant cholesterol (mg/dL)	16.0 (12.0–23.0)	16.0 (11.0–23.0)	16.0 (12.0–23.0)	0.0012	0.0013	0.3382
Metabolic syndrome (%)	25.42% (15,509)	25.65% (11,727)	25.66% (2148)	0.0211	0.0124	0.0891
*APOB* rs679899	AA (77,368)	AG (26,778)	GG (2341)			
Total cholesterol (mg/dL)	192.0 (171.0–216.0)	195.0 (172.0–218.0)	198.0 (175.0–223.0)	0.0052	0.0005	6.32 × 10^−29^
LDL cholesterol (mg/dL)	119.0 (99.0–140.0)	121.0 (101.0–142.0)	123.0 (103.0–145.0)	0.0080	0.0007	2.50 × 10^−30^
Non-HDL cholesterol (mg/dL)	137.0 (115.0–161.0)	140.0 (118.0–164.0)	143.0 (120.0–168.0)	0.0084	0.0006	1.93 × 10^−40^
HDL cholesterol (mg/dL)	53.0 (45.0–63.0)	53.0 (45.0–63.0)	53.0 (45.0–63.0)	−0.0030	0.0006	6.71 × 10^−8^
Triglyceride (mg/dL)	90.0 (64.0–133.0)	92.0 (64.0–133.0)	96.0 (65.0–138.0)	0.0074	0.0013	1.23 × 10^−8^
Remnant cholesterol (mg/dL)	16.0 (11.0–23.0)	17.0 (12.0–24.0)	17.0 (12.0–25.0)	0.0116	0.0016	2.47 × 10^−13^
Metabolic syndrome (%)	25.32% (21,139)	26.08% (7579)	26.03 (666)	0.0488	0.0154	0.0016
*APOB* rs13306194	GG (77,614)	GA (26,682)	AA (2191)			
Total cholesterol (mg/dL)	194.0 (172.0–218.0)	190.0 (169.0–213.0)	185.0 (164.0–208.0)	−0.0100	0.0005	2.76 × 10^−101^
LDL cholesterol (mg/dL)	120.0 (100.0–142.0)	117.0 (98.0–137.0)	114.0 (94.0–134.0)	−0.0140	0.0007	5.13 × 10^−89^
Non-HDL cholesterol (mg/dL)	139.0 (117.0–163.0)	135.0 (114.0–158.0)	130.0 (110.0–151.0)	−0.0145	0.0006	2.65 × 10^−115^
HDL cholesterol (mg/dL)	53.0 (45.0–63.0)	53.0 (45.0–63.0)	54.0 (46.0–64.0)	0.0014	0.0006	0.0137
Triglyceride (mg/dL)	91.0 (64.0–135.0)	89.0 (63.0–130.0)	85.0 (61.0–122.0)	−0.0143	0.0013	1.10 × 10^−27^
Remnant cholesterol (mg/dL)	17.0 (12.0–23.0)	16.0 (11.0–23.0)	15.0 (11.0–22.0)	−0.0158	0.0016	5.22 × 10^−23^
Metabolic syndrome (%)	25.80% (21,733)	24.92% (7109)	23.44% (542)	−0.0690	0.0159	1.40 × 10^−5^
*APOB* rs1367117	GG (80,541)	GA (24,135)	AA (1811)			
Total cholesterol (mg/dL)	192.0 (171.0–216.0)	195.0 (173.0–219.0)	200.0 (176.0–225.0)	0.0070	0.0005	1.00 × 10^−45^
LDL cholesterol (mg/dL)	119.0 (99.0–140.0)	122.0 (101.0–143.0)	125.0 (103.0–148.0)	0.0109	0.0007	5.52 × 10^−50^
Non-HDL cholesterol (mg/dL)	137.0 (115.0–161.0)	140.0 (118.0–165.0)	145.0 (121.0–171.0)	0.0105	0.0007	4.12 × 10^−56^
HDL cholesterol (mg/dL)	53.0 (45.0–63.0)	53.0 (45.0–63.0)	53.0 (45.0–63.0)	−0.0020	0.0006	0.0006
Triglyceride (mg/dL)	90.0 (64.0–133.0)	91.0 (64.0–134.0)	95.0 (65.0–140.0)	0.0048	0.0014	0.0004
Remnant cholesterol (mg/dL)	16.0 (11.0–23.0)	17.0 (12.0–23.0)	17.0 (12.0–25.0)	0.0070	0.0017	2.76 × 10^−5^
Metabolic syndrome (%)	25.42% (22,094)	25.75% (6742)	27.41% (548)	0.0362	0.0163	0.0258

Abbreviation: SE, standard deviation. Other abbreviations and participant recruitment as in Table 1. Level is presented as median (interquartile range). *p* value: adjusted for sex, age, BMI, and current smoking.

**Table 3 ijms-23-14963-t003:** Stepwise linear regression analysis: lipid profiles.

	Total Cholesterol				LDL Cholesterol				Non-HDL Cholesterol			
	Beta	SE	R^2^	*p* Value	Beta	SE	R^2^	*p* Value	Beta	SE	R^2^	*p* Value
Age (years)	0.0013	0.00002	0.034	<10^−307^	0.0014	0.00003	0.0168	<10^−307^	0.0019	0.00003	0.0352	<10^−307^
Sex (male vs. female)	0.014	0.0005	0.0047	5.87 × 10^−171^	-	-	-	-	−0.0052	0.0008	0.0007	7.93 × 10^−12^
Body mass index (kg/m^2^)	0.0016	0.0001	0.0053	1.49 × 10^−133^	0.005	0.0001	0.0257	<10^−307^	0.0061	0.0001	0.0485	<10^−307^
Current smoking (%)	-	-	-	-	−0.0029	0.0009	0.0001	0.0014	0.0031	0.0009	0.0001	0.0005
*APOB* rs144467873 (GG vs. GA vs. AA)	0.0759	0.0041	0.0033	8.24 × 10^−76^	0.1302	0.0062	0.0045	4.36 × 10^−99^	0.1079	0.0056	0.0036	9.82 × 10^−84^
*APOB* rs13306194 (GG vs. GA vs. AA)	−0.0093	0.0005	0.0041	2.11 × 10^−85^	−0.0128	0.0007	0.0035	2.04 × 10^−73^	−0.0133	0.0006	0.0045	2.07 × 10^−96^
*APOB* rs1367117 (GG vs. GA vs. AA)	0.0047	0.0005	0.0008	3.37 × 10^−21^	0.0075	0.0007	0.0009	6.84 × 10^−24^	0.0072	0.0007	0.0010	5.60 × 10^−27^
	**HDL Cholesterol**				**Triglyceride**				**Remnant Cholesterol**			
Age (years)	0.0001	0.00003	0.0002	7.03 × 10^−9^	0.003	0.0001	0.0194	<10^−307^	0.0050	0.0001	0.0418	<10^−307^
Sex (male vs. female)	0.0623	0.0007	0.0863	<10^−307^	−0.0566	0.0016	0.0199	4.78 × 10^−285^	−0.0149	0.0019	0.0005	5.84 × 10^−15^
Body mass index (kg/m^2^)	−0.0094	0.0001	0.1645	<10^−307^	0.0221	0.0002	0.1505	<10^−307^	0.0100	0.0002	0.0231	<10^−307^
Current smoking (%)	−0.0112	0.0008	0.0014	5.00 × 10^−44^	0.0411	0.0019	0.0036	5.95 × 10^−107^	0.0319	0.0023	0.0033	3.01 × 10^−44^
*APOB* rs1318006 (AA vs. AG vs. GG)	−0.0064	0.0006	0.0008	2.33 × 10^−25^								
*APOB* rs35131127 (TT vs. TC vs. CC)					0.017	0.0017	0.0010	1.09 × 10^−24^				
*APOB* rs13306194 (GG vs. GA vs. AA)					−0.0125	0.0013	0.0007	2.66 × 10^−21^	−0.0138	0.0016	0.0009	2.82 × 10^−17^
*APOB* rs144467873 (GG vs. GA vs. AA)									−0.1048	0.0143	0.0004	2.32 × 10^−13^
*APOB* rs56213756 (CC vs. CG vs. GG)									0.0124	0.0019	0.0007	3.70 × 10^−11^
*APOB* rs679899 (AA vs. AG vs. GG)									0.0060	0.0017	0.0001	0.0006

Participant recruitment criteria for analysis are displayed in Figure 1. Abbreviation as in Table 2.

**Table 4 ijms-23-14963-t004:** Logistic regression analysis: metabolic syndrome.

	Beta	SE	OR (95% CI)	*p* Value
Age (years)	0.0870	0.0185	1.0909 (1.0519–1.1312)	2.70 × 10^−6^
Sex (male vs. female)	0.0664	0.0008	1.0687 (1.0670–1.0704)	<10^−307^
Body mass index (kg/m^2^)	0.3127	0.0025	1.3672 (1.3605–1.3738)	<10^−307^
Current smoking (%)	0.2767	0.0214	1.3188 (1.2646–1.3753)	3.14 × 10^−38^
*APOB* rs144467873 (GG vs. GA vs. AA)	0.6564	0.1162	1.9279 (1.5354–2.4209)	1.60 × 10^−8^
*APOB* rs4665709 (GG vs. GA vs. AA)	0.1189	0.0212	1.1263 (1.0804–1.1742)	2.16 × 10^−8^

Participant recruitment criteria for analysis is displayed in Figure 1. Abbreviation: OR, odds ratio; CI: confidence interval.

**Table 5 ijms-23-14963-t005:** Summary of coefficients used for standard Mendelian randomization analysis: low-density lipoprotein cholesterol (LDL-C) levels and diabetes mellitus (DM).

T_A_	T_B_	G_A_	T_A_-T_B_			G_A_-T_A_			G_A_-T_B_			IV_A_-T_B_			IV_A_-T_B_-adjT_A_		
			Beta	SE	*P ^a^*	Beta	SE	*P ^a^*	Beta	SE	*P ^a^*	Beta	SE	*P*	Beta	SE	*P ^b^*
LDL-C level	DM	*APOB* rs144467873	−2.3131	0.0988	3.25 × 10^−121^	0.1373	0.0061	1.96 × 10^−110^	0.1182	0.2095	0.5726	0.8691	1.5404	0.5726 *^a^* (0.7473 *^c^*)	2.8639	1.5556	0.0656
		*APOB* rs13306194	−2.3131	0.0988	3.25 × 10^−121^	−0.0140	0.0007	5.13 × 10^−89^	0.0561	0.0233	0.0158	−4.0072	1.6611	0.0158 *^a^* (0.0002 *^c^*)	−1.8901	1.6720	0.2583
		*APOB* rs1367117	−2.3131	0.0988	3.25 × 10^−121^	0.0109	0.0007	5.52 × 10^−50^	−0.0534	0.0256	0.0368	−4.8537	2.3245	0.0368 *^a^* (0.0272 *^c^*)	−2.6217	2.3389	0.2623
		WGRS_*APOB*_3SNPs	−2.3131	0.0988	3.25 × 10^−121^	0.8796	0.0277	5.81 × 10^−221^	−2.1620	1.0200	0.0340	−2.4568	1.1591	0.0340 *^a^* (0.0029 *^c^*)	−0.3710	1.1688	0.7509
		WGRS_*APOB*_2SNPs	−2.3131	0.0988	3.25 × 10^−121^	0.8805	0.0375	9.08 × 10^−122^	−3.6711	1.2769	0.0040	−4.1717	1.4511	0.0040 *^a^* (8.2 × 10^−5^ *^c^*)	−2.0614	1.4608	0.1582

WGRS_*APOB*_3SNPs: WGRSs of 3 *APOB* variants *APOB* rs144467873, *APOB* rs13306194, *APOB* rs1367117; WGRS_*APOB*_2SNPs: WGRSs of 2 *APOB* variants *APOB* rs13306194, *APOB* rs1367117. T_A_ and T_B_: phenotypes A (LDL-C level) and B (DM); G_A_: genotypes determining T_A_; IV_A_: instrumental variables for G_A;_ IV_A_-T_B_-adjT_A_: association between IV_A_-T_B_ with adjustment of T_A_ (LDL-C level). *^a^*: Adjustment for age, sex, current smoking status, and BMI; *^b^:* After further adjustment of LDL-C level. *^c^:* Adjustment for age, sex, current smoking status, BMI, and other possible confounders, such as eGFR, platelet counts, hemoglobin, triglyceride, AST, and remnant-cholesterol.

## Data Availability

The data presented in this study are available upon request from the corresponding author.

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
