# Peer review of "Pleiotropic Effects of APOB Variants on Lipid Profiles, Metabolic Syndrome, and the Risk of Diabetes Mellitus"

_ijms, 2022, doi:10.3390/ijms232314963_

Round 1
Reviewer 1 Report
I read and revised the article entitled "Pleiotropic Effects of APOB Variants on Lipid Profiles, Metabolic Syndrome and the risk of Diabetes Mellitus" by Jang et al.
The authors investigated the Taiwanese biobank for common and rare variants of the APOB gene in order to associate them to the development of metabolic syndrome, diabetes mellitus or dyslipidemia.
the authors identified 5 exonic nonsense mutations in APOB and associated them with the development of T2DM, MS, or Hypercholesterolemia.
I found this to be one interesting work with a clear methodology that would be very interesting for publication once revised. Listed beneath you will find major concerns that need to be developed
1 - Statistical analysis. In the methods section, the authors should better explain the statistical test used especially the way stepwise regression was performed.
2- In table 3, it seems that the stepwise regression was made by mixing up all five variants and this is a methodological error that should be revised by your statistician.
2- Overall tables 3 and 5 should be restyled in order to be more comprehensible.
Reviewer 2 Report
The aim of this work is to analyze the significance of APOB variants for lipid profiles and development of metabolic syndrome in the Taiwanese population. Although the results refer to a restricted Asian area, the obtained data are interesting and have a potential for practical application, especially regarding pathological conditions like metabolic syndrome and diabetes mellitus. This is a well written paper with adequate choice of experimental techniques well-performed statistical analysis.
The results are presented in a clear and precise way. The authors provide new original data concerning a part of the Asian population, which is a contribution to the overall picture of the role of APOB for variations in the lipid profile and the metabolic basis of diabetes mellitus.
The interpretation of the presented results in the discussion is sound and well motivated.
In addition, enough recent references are included in the manuscript.
Author Response
We greatly appreciate the comments of the reviewer.
Round 2
Reviewer 1 Report
I have no further comments.